# A Comprehensive Review on Circulating cfRNA in Plasma: Implications for Disease Diagnosis and Beyond

**DOI:** 10.3390/diagnostics14101045

**Published:** 2024-05-17

**Authors:** Pengqiang Zhong, Lu Bai, Mengzhi Hong, Juan Ouyang, Ruizhi Wang, Xiaoli Zhang, Peisong Chen

**Affiliations:** 1Department of Clinical Laboratory, The First Affiliated Hospital of Sun Yat-sen University, Guangzhou 510080, China; 2Department of Pediatrics, The First Affiliated Hospital of Sun Yat-sen University, Guangzhou 510080, China

**Keywords:** cell-free RNA, liquid biopsies

## Abstract

Circulating cfRNA in plasma has emerged as a fascinating area of research with potential applications in disease diagnosis, monitoring, and personalized medicine. Circulating RNA sequencing technology allows for the non-invasive collection of important information about the expression of target genes, eliminating the need for biopsies. This comprehensive review aims to provide a detailed overview of the current knowledge and advancements in the study of plasma cfRNA, focusing on its diverse landscape and biological functions, detection methods, its diagnostic and prognostic potential in various diseases, challenges, and future perspectives.

## 1. Introduction

In recent times, there has been a rapid advancement in liquid biopsy technology. In comparison to conventional tissue biopsy, liquid biopsy technology offers several advantages, including reduced invasiveness, enhanced repeatability, the ability to overcome tumor molecular spatial heterogeneity, and the capacity to dynamically reflect tumor changes [1]. Consequently, it has found extensive application in early cancer screening, guiding patients towards adjuvant therapy, evaluating the effectiveness of chemotherapy, and providing insights into tumor prognosis [2]. Liquid biopsy technology, a highly anticipated early-detection technology for cancer [1,3,4], encompasses markers such as circulating tumor cells (CTCs), extracellular DNA (cfDNA) and its methylation, extracellular cell-free RNA (cfRNA), and exosomal proteins [5]. Among the various liquid biopsy samples, plasma cell-free DNA (cfDNA) and plasma cfRNA in peripheral blood have garnered significant attention. These DNA or RNA molecules are products of normal cell metabolism and apoptosis, as well as tumor cell activity, and they hold substantial value for clinical applications.

cfRNA, also known as extracellular RNA (exRNA), is found in bodily fluids such as blood, urine, alveolar lavage fluid, and pleural fluid. As showed in Figure 1, it primarily consists of various types of RNA, including messenger RNA (mRNA), small RNA (miRNA), long non-coding RNAs (lncRNAs), and circular RNA (circRNA). Circulating free RNA is the predominant form of free RNA existing in blood. The exact source of cfRNA remains uncertain. Zhou et al. conducted an in vitro study which revealed that the concentration of exRNA can increase under conditions of hypoxia and increased cell metabolism [6]. This finding may offer a possible explanation for the elevated levels of cfRNA observed in cancer patients. However, this has not yet been validated through in vivo testing. cfRNA originates from cellular activities such as apoptosis, necrosis, or active secretion, and can be released from the nucleus, cytoplasm, or extracellular vesicles (e.g., exosomes). Circulating cfRNA, released into the extracellular environment and present in body fluids such as plasma, holds immense promise as a non-invasive source of genetic information. The analysis of plasma cfRNA offers a unique opportunity for early disease detection, monitoring treatment responses, and understanding disease progression.

As a biomarker, cfRNA is sensitive and functional. The expression of many RNAs, including those from human and microbial sources, is tissue-specific [7]. Changes in specific RNA expression profiles in different tumors can be reflected in plasma. Therefore, biomarkers based on cfRNA not only have many signals in patient plasma at the early stage of cancer [8], but also provide more functional information and can be used as an early diagnostic tool for a variety of diseases, including cancer [9]. Plasma cfRNA is derived from many tissues in the body [10], and its detection can help in indirectly observing pathogeneses in real-time and identifying physiological changes associated with the prephase of subtypes. RNA can more actively enter the environment outside of the cell through cellular efflux mechanisms, such as exosomes [11]. This aspect is particularly significant as it allows for liquid biopsies to serve as a potential means of indirectly observing the pathogenesis in real time and identifying physiological changes associated with the prephase of subtypes. Moreover, effective cfRNA sequencing targeting cancer signals has a lower cost than cfDNA and its methylation sequencing, making it more beneficial to the popularization of early screening and early diagnosis [12]. In addition, compared with cfDNA, cfRNA analysis provides more valuable information on gene expression, splicing, and post-transcriptional regulation. Under specific conditions, such as inflammation, an immune response caused by microbial infection or a tumor, changes in the microenvironment will produce plenty of specific cfRNA molecules [7].

Despite the advancements in high-throughput sequencing instruments, library preparation techniques, and bioinformatics pipelines documented over the last 15 years, and the evidence showing that cfRNA has great potential in cancer diagnosis [13,14], treatment [15], and personalized medicine [16], the implementation of cfRNA diagnostics in clinical environments remains limited. This can be attributed, at least in part, to the absence of a singular dominant technology that effectively, easily, and economically tackles all of the associated obstacles. In this review, we discuss the current knowledge and advancements in the study of plasma cfRNA, focusing on detection methods, its diagnostic and prognostic potential in various diseases, its biological functions, challenges, and future perspectives (Figure 1).

## 2. Diverse Landscape of Plasma cfRNA

Compared with urine, cerebrospinal fluid (CSF), and other body fluids, blood encompasses a rich composition of both universal and specialized characteristics. Its universal characteristics involve the facilitation of fluid transport within the circulatory system [17], the regulation of body temperature [18,19] and acid–base balance [20], and the transportation of oxygen and carbon dioxide [21,22]. Additionally, blood plays a crucial role in immune responses [23], demonstrating its essential function in maintaining physiological homeostasis. On the other hand, blood’s particularity includes specific cellular components, such as red blood cells, white blood cells, and platelets, each with unique functions. The red blood cells are designated for carrying oxygen, while leukocytes are engaged in immune responses, and platelets contribute significantly to the coagulation process. Another distinctive feature is the presence of plasma, which is the liquid component of blood containing water, electrolytes, and proteins [24,25]. Consequently, it plays a crucial role in nutrient and hormone transportation, as well as in the process of blood coagulation. Additionally, blood has a special coagulation mechanism allowing it to rapidly form blood clots in response to an injury or vascular damage, thanks to the involvement of coagulation factors, platelets, and related components. Lastly, blood is involved in nutrient metabolism by transporting nutrients such as glucose and amino acids, as well as by eliminating metabolites like urea and lactic acid, thereby participating in the overall nutrient metabolism of the body [26,27]. Blood is universal in terms of its circulation, oxygen transport, and immune defense. However, its unique cell composition, clotting mechanism, and nutrient transport function result in particularities that distinguish it from other body fluids. As a result, blood plays a distinctive and irreplaceable role in maintaining overall physiological balance and responding to external changes.

The plasma cfRNA landscape includes a diversity of RNA molecules, detection methods, and clinical roles, and covers various diseases. The diversity of cfRNA is characterized by the presence of various RNA molecules in the bloodstream, encompassing a range of RNA types and functions. Notably, this diversity constitutes a crucial attribute of cfRNA as a potential biomarker. One type of RNA found in plasma is messenger RNA (mRNA), which originates from different cells and holds information about the expression of specific genes. As a result, mRNA in plasma has the potential to serve as a valuable indicator of cellular activity in disease states and therapeutic responses [28]. Another class of RNA found in plasma is miRNA, typically composed of 20–22 nucleotides. miRNA plays a key role in regulating gene expression and can impact the level of gene expression by targeting specific mRNA. Therefore, miRNA present in plasma can be utilized as a potential marker for disease diagnosis and prognosis [29]. Additionally, long non-coding RNAs (lncRNAs), a type of long RNA molecule that does not code for proteins, are involved in the regulation of gene expression and cellular function. Certain lncRNAs detected in plasma are associated with the development and progression of diseases, such as tumors and cardiovascular diseases [30,31]. Another distinctive type of RNA present in plasma is circular RNA (circRNA), a special closed-loop RNA molecule. CircRNA found in plasma is believed to be closely linked to the occurrence and development of various diseases, particularly tumors [32]. Furthermore, several other types of RNA, including piRNA [33], snRNA [34], and snoRNA [35], have been identified in plasma. Although ongoing research is investigating their functionality and effects, these types of RNA may hold significant implications for the diagnosis and treatment of specific diseases. Collectively, the diverse array of cfRNA molecules reflects the intricate regulation of gene expression in cellular activity and disease states. Realizing their potential value in disease diagnosis, prognostic evaluation, and therapeutic monitoring depends on gaining an in-depth understanding of their diversity and functionality. Thus, further research is essential to discern their specific applications and significance in clinical practice.

## 3. Biological Functions of cfRNA

As showed in Figure 1, cfRNA represents a complex and actively researched area within biology. These RNA molecules exist as fragments of varying lengths and are associated with multiple potential biological functions. Firstly, cell-to-cell communication may occur via cfRNA, which acts as a vector transmitting RNA fragments through the bloodstream to affect the biological processes of distant cells. Extensive research has shown that cfRNA can be transmitted between cells through mechanisms such as urinary bubble RNA-binding proteins, which serve as carriers of information. This intercellular communication serves the purpose of regulating cellular function at locations far from where the cells were produced, such as during damage repair, disease states, and immune responses [36,37]. However, the exact mechanisms and influence of cfRNA in intercellular communication remain subject to ongoing investigation. Therefore, further experiments and studies are necessary to uncover its detailed molecular mechanisms and biological function.

Secondly, plasma cfRNA can directly participate in gene regulation in target cells, with miRNA, for example, inhibiting gene translation or inducing gene degradation by binding to the mRNA of target genes, thereby impacting the functional immune regulation of the target cells [38,39]. The gene regulation function of cfRNA is an actively evolving field of research, characterized by continuous discoveries and insights. Plasma cfRNA molecules serve a critical role in intercellular communication and gene regulation. miRNA, for example, acts through a mechanism wherein it binds to the mRNA of target genes, leading to the inhibition of gene translation or the promotion of mRNA degradation [40]. Exosome bubbles or binding proteins facilitate the release of RNA into the extracellular environment, where they exert a regulatory influence on distant cells. The interactions of RNAs with target genes in plasma play a pivotal role in various biological processes such as cell proliferation [41], apoptosis [42], differentiation [39], metabolism [43], and immune responses [44], thereby significantly impacting disease pathogenesis and progression. On the other hand, long non-coding RNAs (lncRNAs) are characterized as long-length non-coding RNAs that impact gene expression through various regulatory mechanisms, including chromatin modification [45] and transcriptional regulatory protein stability [46]. Similar to other RNAs, lncRNAs can be transferred to other cells in plasma through extracellular vesicles, such as exosomes, to exert a distant regulatory influence. Notably, lncRNA is instrumental in tumor development, the regulation of the cell cycle, and other key cellular processes [46,47]. Additionally, small RNA fragments present in plasma may regulate protein translation by binding to mRNA, thereby influencing gene expression. These small RNA fragments are capable of being transferred between cells, affecting cellular function by regulating gene expression levels [48,49]. Ultimately, the regulatory functions of cfRNA molecules have far-reaching implications in various physiological and pathological processes, underpinning their significance in cellular and molecular biology research.

Furthermore, several studies indicate that cfRNA may influence immune system responses by modulating the activity of immune cells, thereby affecting immune responses to combat infection and control tumors [50,51]. cfRNA has an immunomodulatory function that impacts the regulation of immune cells. miRNA in plasma can be released into the environment through exosomes or bubbles and taken up by other cells, including immune cells [52]. Once internalized, RNAs have the potential to regulate the expression of target genes, affect the development and differentiation of immune cells, and inhibit inflammation. Some RNAs are particularly important in inhibiting the inflammatory response by reducing the inflammatory response and maintaining the immune system balance through the regulation of inflammation-related genes [53,54]. Furthermore, plasma miRNA’s effects on the immune system extend to T-cell regulation, wherein it may influence the antiviral immunity by regulating the function of T-cells, including promoting or inhibiting their proliferation, differentiation, and apoptosis [55,56,57]. Certain RNAs may play a key role in regulating antiviral immunity by influencing the expression level of antiviral genes and immune cells’ responses to viruses, thereby affecting the overall function of the immune system by regulating the expression of immunoregulatory factors, such as interferon chemokines [58,59]. In addition, cfRNA can impact the polarization of immune cells, such as M1/M2 macrophage polarization, thereby affecting the balanced immune tolerance of inflammatory responses and immune regulation [60]. Furthermore, studies have indicated that cfRNA molecules are associated with T-cell tolerance by affecting T-cell development and function, regulating the immune system’s tolerance to autoantigens, and preventing the occurrence of autoimmune diseases [61]. Abnormal expressions of cfRNA are linked to the occurrence and development of autoimmune diseases, such as rheumatoid arthritis [62] and systemic lupus erythematosus [63].

Additionally, certain cfRNA molecules may also be involved in apoptosis and cell survival, which are crucial for tissue homeostasis. The regulation of apoptosis by miRNA is a multifaceted process that encompasses both the promotion and inhibition of cell death. Some RNAs have been identified as proponents of apoptosis, achieving this by either inhibiting anti-apoptosis genes or augmenting the expression of apoptosis-inducing factors [64]. These specific RNAs are present in plasma and may be transported to target cells through exosomes or other carriers, resulting in a direct impact on cell survival. Moreover, the interaction of RNAs with apoptosis-related genes such as Bcl-2 [65] and Caspases [66] serves to modulate the expression levels of these genes, subsequently influencing the susceptibility of cells to apoptosis. Conversely, there are RNAs that exhibit anti-apoptotic properties, exerting their effect by promoting the expression of anti-apoptosis genes or hampering the activity of apoptosis-inducing factors to sustain cell survival. This dichotomy highlights the role of cfRNA in delicately regulating the balance between cell survival and apoptosis [67]. Additionally, cfRNA can influence cell survival states through the modulation of survival signaling pathways, including the PI3K/AKT and MAPK pathways [68,69]. Beyond miRNA, other small RNA fragments in plasma may also partake in the regulation of cell survival and apoptosis by interacting with mRNA or proteins to modulate intracellular signaling pathways. This underscores the intricate nature of miRNA and other small RNA fragments in orchestrating the delicate intertwined processes of cell survival and apoptosis regulation [70,71,72].

Additionally, alterations in cfRNA may serve as potential biomarkers for diseases, notably apoptosis and survival in cancer and other diseases, reflecting biological changes in cells and tissues. Moreover, some studies have shown that they may regulate pathways related to angiogenesis [73], impacting the development and maintenance of the vascular system, as well as playing a regulatory role in neurodevelopmental, neuroprotection, and neurodegenerative diseases [74,75]. Importantly, it is crucial to recognize that the specific function of cfRNA may vary depending on its species, and different types of RNA may play distinct roles in the body. Consequently, ongoing research is imperative to uncover the numerous unknowns and fully understand the biological function of cfRNA, given its complex nature and the need for thorough and continuous investigation.

## 4. Methods of cfRNA Detection

Various techniques have been developed to detect and analyze cfRNA in plasma. The detection of cfRNA involves several methods, beginning with RNA extraction as the initial step. Due to the low concentration of RNA in plasma, efficient extraction methods such as the phenol/chloroform silica gel column and magnetic bead methods are commonly employed to ensure sufficient RNA yields. Following extraction, reverse transcription (RT) is essential to convert the single-stranded RNA in plasma into complementary DNA (cDNA), a key step in the detection process. Common RT methods include reverse transcription polymerase chain reaction (RT-PCR) and reverse transcriptase chain amplification (RT-qPCR). Subsequently, quantitative PCR, including real-time fluorescent quantitative PCR (qPCR), is employed to amplify and detect specific RNA molecules, enabling the quantification of mRNA and miRNA in plasma. Additionally, the traditional method of Northern Blotting, though less commonly used than PCR techniques, remains valuable in specific instances for separating RNAs via electrophoresis and detecting specific RNAs via membrane transfer and probe hybridization. Next-generation sequencing (NGS) technology provides comprehensive information about different RNA species and their relative abundance in plasma, thereby facilitating the discovery of new RNA markers and understanding of the overall diversity of cfRNA [76]. Furthermore, flow cytometry, a cytological technique, can rapidly and accurately quantify specific cfRNA molecules by combining them with a fluorescently labeled probe [77]. When selecting the detection method, consideration must be given to the sensitivity, specificity, and availability of laboratory technology and equipment. In general, a combination of detection approaches may provide a more holistic understanding of the diversity and abundance of free RNA in plasma.

As showed in Table 1, various methods are used for the detection of cfRNA, each with distinct advantages and limitations. Real-time fluorescence quantitative PCR (RT-qPCR) is a widely utilized method due to its high sensitivity, good quantification, and suitability for the detection of specific RNA sequences [78]. However, it is limited to the detection of known RNA sequences and requires prior knowledge of the target RNA’s sequence information [79,80]. Next-generation sequencing (NGS) provides the ability to comprehensively detect RNA sequencing (RNA-seq) in plasma with high throughput, including known and unknown sequences, making it suitable for the discovery of new RNA markers and offering detailed transcriptomic information [10,81]. Despite these advantages, it comes with relatively high costs and data analysis difficulties, particularly in large-scale epidemiological studies [76,82]. Polyadenylation ligation-mediated sequencing (PALM-Seq), also known as RNA sequencing, is a technique used to study RNA molecule sequences and is often associated with the presence of Polyadenosine (PolyA) in RNA. This technique offers numerous advantages and disadvantages that are important for researchers to consider. PALM-Seq offers several advantages, such as the ability to quantify transcription and discover new transcripts, as well as the ability to detect variable splicing events and study RNA modifications. However, it is important to consider the associated drawbacks, including the high cost and stringent RNA quality and integrity requirements. Furthermore, PALM-Seq does not provide absolute gene expression levels and necessitates the use of complex bioinformatics tools and algorithms for data analysis [83,84]. Digital PCR, specifically droplet digital PCR (ddPCR), is a widely used technology for achieving more accurate detection. It falls under the third-generation PCR category, and it offers four primary application directions, including absolute quantification [85], rare copy detection [86], copy number variation analysis [87], and the determination of gene expression levels [88]. ddPCR provides more accurate quantitative information, particularly for low-copy-number RNA detection, making it suitable for rare events [89,90]. On the other hand, mass spectrometry can be used to detect a variety of RNA modifications and provide detailed RNA structure information [91,92], albeit with the requirement of complex instruments, professional techniques, high costs, and a limited scope of application [93,94]. Flow cytometry can be applied to detect cfRNA, which includes the analysis of RNA expressions of specific cell subsets. This technique can also be combined with appropriate antibodies or fluorescent probes to identify modifications on RNAs, such as m6A (N6-methyadenylate), as well as for investigating intracellular RNA localization and dynamics [95]. Flow cytometry is primarily advantageous in the detection of cell surface markers and proteins, whereas its applications in RNA detection are somewhat limited. When compared to other RNA analysis techniques like RT-qPCR or RNA sequencing, flow cytometry may present some drawbacks, particularly for complex RNA expression analyses [43,96]. Furthermore, the limitations of RNA distribution in tissues highlight the need for suitable methods to study such distributions, as the detection of free RNA in plasma may not be sensitive enough and is more appropriate for tissue sample selection. Ultimately, selecting the appropriate method depends on the specific purpose of the study, the budget, the sample size, and the available laboratory equipment and technology. Combining multiple methods can provide a more comprehensive understanding of free RNA in plasma. We summarized the methods of cfRNA detection in recent research (Table 2).

## 5. Circulating cfRNA in Disease Diagnosis

Plasma cfRNA has shown great promise as a diagnostic tool in various diseases. In cancer, cfRNA analysis enables the detection of tumor-specific genetic alterations, facilitating early diagnosis and the monitoring of treatment responses [81]. In certain instances, cancer cells release specific RNA or miRNA molecules that can be identified in plasma. This makes cfRNA a potential marker for early cancer diagnosis, even in small tumors without clear symptoms [76]. The analysis of free RNA has emerged as a valuable tool with significant potential in cancer diagnosis, cancer detection, the prediction of tumor origin tissues, and the identification of cancer subtypes [97,98]. This promising analyte offers a unique opportunity to uncover tissue- and subtype-specific biomarkers in breast and lung cancer patients [101]. It is essential to establish a baseline cell-free transcriptome in the absence of cancer, facilitating the identification of tissue specificity and subtype specificity in breast and lung cancer patients [16]. After cancer treatment, monitoring changes in cfRNA can help detect cancer recurrence earlier. Regular testing of plasma samples from patients can lead to the timely treatment of potential recurrences. One study found that plasma miR-4442 levels were associated with colorectal cancer (CRC) recurrence and showed an incremental increase with earlier recurrence dates. Furthermore, miR-4442 demonstrated high sensitivity and specificity as a potential biomarker for early CRC recurrence. Subsequent analyses indicated that the expression of miR-4442 in cancer tissues of patients with metastatic liver cancer from CRC was higher than that in normal liver, CRC, and normal colorectal tissues. Notably, the overexpression of miR-4442 promoted the proliferative, migratory, and invasive activities of CRC cells, while also resulting in decreased levels of RBMS1 and E-cadherin, and increased levels of N-cadherin and Snail1. These findings underscore the clinical utility of plasma miR-4442 as a biomarker for predicting the early recurrence of CRC [104]. In addition, cfRNA has shown to have a potential biomarker role in a variety of cancers. The study of cfRNA has made significant progress in the field of breast cancer, and several miRNA and long-stranded RNA molecules in cfRNA have been found to be abnormally expressed in breast cancer patients [105,106]. These RNAs have the potential for use in the diagnosis of early breast cancer, prediction of patient prognoses, and monitoring of treatment responses [107,108]. Furthermore, changes in the expression levels of specific miRNA molecules in the plasma of lung cancer patients have been linked to the occurrence and development of lung cancer, indicating potential clinical applications for early diagnosis [102,103]. Additionally, some miRNA and mRNA expression levels in plasma exhibit significant variation in patients with gastric cancer [109,110] and liver cancer [111,112], presenting promising prospects for early diagnosis and treatment monitoring of these cancers. Similarly, research focusing on prostate cancer has been seeking potential biomarkers to aid in diagnostic and therapeutic monitoring [113,114], while studies on colorectal cancer have indicated associations between certain miRNA molecules in plasma and the development and progression of this disease [7,115]. Moreover, scientists are actively exploring the potential application of cfRNA as a biomarker in other cancer types, emphasizing the need for further in-depth research and validation in this area. It is important to note that despite promising research results, the application of these markers in clinical practice still requires additional validation and standardization.

In infectious diseases, the presence of pathogen-specific RNAs in plasma cfRNA can aid in the identification of the causative agent [116,117]. Furthermore, cfRNA analysis has potential applications in cardiovascular diseases [47,118,119] and neurodegenerative conditions [120]. The detection and analysis of cfRNA as a potential biomarker in disease diagnosis can provide valuable insights into the development and treatment responses of diseases. Changes in the expression levels of cfRNA are associated with the onset and progression of various diseases, including cancer, cardiovascular diseases, infectious diseases, and neurological diseases. Identifying these changes enables the identification of potential biomarkers for early diagnosis and disease monitoring in diverse conditions. Notably, in cancer diagnosis, cfRNA released by tumor cells can be detected in the blood, making it a non-invasive cancer marker. Studies have explored the potential application of cfRNA in various cancer types, such as breast cancer, lung cancer, and gastrointestinal tumors. Moreover, the analysis of cfRNA in detecting infectious diseases, such as viral infections, has potential clinical applications for early diagnosis and monitoring [99]. Additionally, the alteration of neurosystem-related RNAs in neurological diseases, such as Alzheimer’s [121] and Parkinson’s disease [122,123], provides potential biomarkers for diagnosis and therapeutic response monitoring. It is essential to acknowledge that while cfRNA holds potential in disease diagnosis, further research and validation are needed. Consideration of the sensitivity and specificity of the technology available is crucial to ensure its reliability in clinical practice as this field continues to evolve. 

## 6. Circulating cfRNA in Disease Prognosis

cfRNA plays a crucial role in disease prognosis assessments, serving as a prognostic marker, monitoring treatment responses, and providing insights into disease progression. Specifically, in diseases like cancer, alterations in cfRNA levels can be indicative of disease prognosis. For instance, the types and quantity of tumor-associated free RNA in a patient’s plasma can be linked to the risk of disease progression and recurrence, as well as the overall prognosis [60]. Furthermore, monitoring changes in cfRNA during treatment can offer valuable information regarding a treatment’s effectiveness and a disease’s progression. Treatment-induced fluctuations in specific RNA levels can be utilized to evaluate a patient’s response to treatment and forecast disease progression rates and severity [2]. By scrutinizing cfRNA, medical professionals can gain a deeper understanding of a patient’s pathophysiological state and disease characteristics, and can tailor individualized treatment plans based on RNA marker variations. Despite the potential of cfRNA in disease prognosis, extensive research and clinical validation are imperative to identify the most predictive RNA molecules for specific diseases and ensure the accuracy and reliability of corresponding detection methods. This comprehensive approach will enable the optimization of treatment strategies and enhance the overall treatment effectiveness.

The examination of cfRNA’s role in disease prognosis, particularly miRNA and other RNA molecules influencing disease therapeutic responses and patient prognoses, is currently a focal point of research. Specific levels of cfRNA are associated with the clinical features, survival, and treatment responses of tumors, presenting a crucial role in tumor prognosis. The analysis of miRNA expression in plasma through survival analysis provides valuable information for patient prognosis, with particular RNAs indicating a favorable prognosis and others signaling a worsening prognosis. Tumor-associated RNAs, found in plasma, are crucial for tumor prognosis. Several studies have linked specific miRNA expression levels to tumor clinical features, survival, and responses to treatment [55,74]. cfRNA also substantially impacts the prognosis of cardiovascular diseases. Changes in the expression of certain RNAs in plasma following myocardial infarction are closely linked to a patient’s prognosis and can predict cardiovascular events, enabling the use of specific cfRNA molecules to forecast the risk of cardiovascular events and facilitate the early intervention and treatment of neurological outcomes [47,119]. Furthermore, alterations in miRNA expression in plasma are associated with the prognosis of neurological diseases and patient outcomes, including the development of neurodegenerative diseases. cfRNA molecules are also biomarkers for assessing a patient’s prognosis post stroke and have the potential to serve as biomarkers for predicting the prognoses of infectious diseases, reflecting immune responses and disease progression [74]. Additionally, cfRNA is correlated with the treatment responses and progression of immune system diseases, offering insights into predicting long-term patient outcomes. The role of cfRNA in disease prognosis spans various categories, including tumors, cardiovascular diseases, neurological diseases, infectious diseases, and immune system diseases [23,46]. Consequently, changes in the expression patterns and levels of these RNA molecules hold the potential to serve as biomarkers for enhancing prognostic assessments of diseases, guiding treatment decisions, and supporting personalized medicine. Nevertheless, additional research and validation are required to elucidate their precise mechanisms and specific clinical applications.

## 7. Circulating cfRNA in Disease Treatment

The potential role of cfRNA in disease treatment encompasses several crucial aspects. Firstly, certain expression patterns of cfRNA bear substantial implications for the prognosis and therapeutic response of specific diseases. This analysis aids in the prediction of disease progression, formulation of treatment plans, and assessment of treatment effectiveness. Moreover, alterations in cfRNA during treatment can function as a biomarker of therapeutic effects, affording valuable insights into therapeutic responses and guiding the adaptation of treatment protocols. Importantly, the potential utility of cfRNA extends beyond diagnosis, serving as a tool to identify new therapeutic targets linked to diseases [100]. Systematic analyses of cfRNA can yield information conducive to the discovery of novel drugs or treatments that enhance the efficacy of disease-specific treatments. Furthermore, the examination of cfRNA contributes to an enhanced understanding of disease biology. By scrutinizing the expression patterns and function of these RNA molecules, researchers can garner insights into the biological mechanisms of diseases, thereby informing the development of more efficacious treatment strategies [15,106]. 

cfRNA plays a pivotal role in therapeutic monitoring, fulfilling various functions in this process. Firstly, the evaluation of therapeutic responses relies on the expression pattern or level of cfRNA, which may change as treatment progresses. By regularly testing cfRNA, doctors can assess a patient’s response to treatment. In cases where treatment is effective, the RNA associated with a particular disease may decrease or disappear, while its expression may increase if treatment is less effective or if the disease relapses. Consequently, this monitoring aids in tracking treatment effectiveness and disease progression. Moreover, cfRNA analysis facilitates the early prediction of treatment effects. Specifically, it allows doctors to anticipate a treatment’s efficacy in the initial stage, offering insights into whether the patient has positively responded to the treatment. This early knowledge enables adjustments to the treatment plan, thereby enhancing the likelihood of treatment success. Notably, individual variations in patients’ responses to the same treatment due to differing pathophysiological characteristics make it essential to monitor cfRNA [19]. This approach enables doctors to grasp the patient’s condition better and tailor their treatment plan based on individual differences, thereby enhancing treatment targeting and effectiveness. Furthermore, monitoring cfRNA serves as a useful way to detect disease recurrence after treatment completion. Some diseases may experience remission post treatment, and continuous monitoring of RNA changes helps doctors identify signs of disease recurrence promptly. This timely detection enables necessary measures to be taken to mitigate disease resurgence. Lastly, the monitoring of cfRNA minimizes the likelihood of unnecessary treatments being administered for patients unsuitable for or not benefiting from a specific treatment. By adjusting treatment plans based on monitoring results, doctors can better manage patients’ treatment, reducing unnecessary side effects and costs [95].

cfRNA plays an important role in the discovery of drug targets and new therapies, as reflected in several key aspects. Firstly, changes in cfRNA are closely related to the development and progression of diseases, making it possible to identify biomarkers associated with specific disease states through the analysis of free RNA in the plasma. These biomarkers can serve as potential drug targets or indicators of therapeutic responses, facilitating a better understanding of the molecular mechanisms of diseases [124]. Additionally, studying the abnormal expression of free RNA in disease states enables researchers to identify key molecules associated with disease progression that may become new drug targets. This understanding helps in developing more targeted and effective pathophysiological drug mechanisms [41]. Furthermore, analyzing the expression profile of cfRNA in different disease states provides insights into the molecular basis of diseases, revealing their key signaling pathways and biological processes. This understanding supports the design of new therapeutic strategies and individualized treatment plans based on the expression patterns of specific RNAs in a patient’s plasma, aiming to improve treatment targeting, reduce side effects, and increase treatment success rates. Lastly, the analysis of cfRNA can be utilized to assess the efficacy and safety of drugs, enabling the full assessment of drug effectiveness and the timely detection of potential adverse reactions or toxicity by monitoring changes in the RNAs in a patient’s plasma during treatment [15,106].

Despite its potential applications in disease treatment, the application of cfRNA is fraught with challenges related to the stability of standardized RNAs obtained from samples, standardized analytical techniques, and clinical validation. Nonetheless, with the continuous advancement of technology and comprehensive research, the potential role of cfRNA as a biomarker in disease treatment will attract considerable attention. We have summarized the clinical value of cfRNA and methods for its detection according to recent research in Table 2.

## 8. Challenges and Future Perspectives

Despite the potential clinical applications of cfRNA in disease diagnosis and prognostic assessments, practical challenges hinder its effective implementation. One such challenge is the sourcing and handling of samples. The content of cfRNA is relatively low and is susceptible to interference from external factors. Hence, ensuring the integrity of RNA molecules requires meticulous attention when selecting the conditions for sample collection, processing, and storage. Improper treatment can result in RNA degradation, undermining the reliability of subsequent analyses. Various types of RNases are present in the blood, such as endonucleases and exonucleases. These enzymes can significantly affect the detection of mRNA. RNase enzymes in the blood are responsible for breaking down free RNA molecules, including mRNA. Consequently, if blood samples become contaminated with RNases during collection, processing, or storage, the mRNA contained within them can undergo rapid degradation, leading to compromised experimental accuracy [125]. To mitigate the detrimental effects of RNases on mRNA integrity, a range of protective measures are typically implemented during RNA sample handling and storage. These precautions may include incorporating RNase inhibitors, storing the samples at low temperatures, and minimizing the samples’ exposure to room temperature. Blood samples should be processed promptly following collection, and precautions should be taken to prevent contact with RNase-containing materials during sample handling in order to minimize RNA sample degradation attributable to RNase activity [126,127].

Moreover, the lack of standardized sample processing and analysis methods presents a hurdle, as different laboratories and research teams employ varied techniques. This diversity complicates result comparison and integration, emphasizing the need for standardized experimental procedures. Furthermore, the biological diversity across populations contributes to significant variations in cfRNA expression levels among individuals and over time. Therefore, large-scale clinical studies must account for these differences to establish biomarkers with reliable specificity and sensitivity. In this context, the criticality of ensuring assay specificity and sensitivity in early disease diagnosis and prognostic assessments becomes evident. The accurate detection of small RNA changes is crucial for precise results. 

When performing RNA sequencing, the fragmented nature of circulating nucleic acids poses a significant challenge, particularly for shorter miRNA and other small non-coding RNA fragments. This nature of fragmentation may potentially affect the efficacy of certain RNA types, such as mRNA and miRNA, as detected with sequencing technology [55,67]. To address this challenge and enhance RNA detection, various strategies can be implemented. These strategies include optimizing sample processing and extraction methods to minimize the fragmentation of circulating nucleic acids. Additionally, utilizing specific RNA extraction kits or methods can help safeguard the integrity of RNA molecules. Selecting appropriate sequencing technologies and analytical methods is crucial for maximizing the sensitivity and accuracy of circulating nucleic acid detection. Moreover, integrating bioinformatics analyses with existing knowledge of RNA structures and functions can aid in the interpretation of sequencing data, leading to the more precise identification and quantification of circulating nucleic acids [35]. Various methods exist for extracting free RNA from plasma or serum samples, as these samples contain free RNA molecules. One common method is the plasma/serum total RNA extraction method, which is known for its versatility in extracting RNAs of different lengths, including small and long RNA molecules. This method boasts a high extraction efficiency, making it suitable for larger sample sizes. Commercial kits designed for this purpose are readily available in the market, enabling the extraction of RNAs suitable for a range of downstream applications, such as RT-qPCR and RNA sequencing. However, the drawbacks of this method include the relatively high costs of the kits, the complexity of the operation process, the requirement for specialized laboratory equipment, and the need for advanced technology. Another specialized method is the miRNA extraction method, specifically tailored for extracting small RNA molecules, such as miRNA, from plasma or serum samples. This method is particularly advantageous for capturing low-abundance miRNA. Nonetheless, it is characterized by its high cost and limitations on the types and lengths of RNAs it can extract, making it unsuitable for the extraction of long RNA molecules [1]. The magnetic bead separation method is also commonly employed due to its use of magnetic bead binding technology, which facilitates easy operation and fast extraction. This method is suitable for extracting RNA molecules of varying lengths from plasma or serum samples. However, it requires the use of special equipment, such as a magnetic bead separator, leading to increased experimental costs. Additionally, some methods may exhibit limitations in terms of the purity and extraction efficiency of the RNA molecules obtained. For those seeking a rapid extraction process, there is the rapid extraction method, characterized by its simplicity and quick extraction of free RNA. This method is ideal for extracting RNAs from small sample volumes, but it may not achieve extraction efficiencies comparable to those achieved with other methods and comes with a higher cost [128]. In summary, each extraction method has its own set of advantages and disadvantages. Researchers must carefully assess their experimental requirements, sample characteristics, and laboratory capabilities to select the most suitable method for their specific needs.

Moreover, interpreting the biological significance of cfRNA and linking it to specific disease states poses a complex challenge, necessitating independent validation of the discovered biomarkers and clinical validation in large-scale studies. Ethical and regulatory considerations add another layer of complexity, encompassing issues such as privacy and lawful sample use. Furthermore, the cost of detecting cfRNA in practical applications may be a limiting factor, underscoring the need for affordable assays for wider clinical use. Notwithstanding these challenges, ongoing scientific efforts are aiming to address these issues, fueling advancements in the application of cfRNA as a biomarker for disease diagnosis and prognosis assessments. With advancing technology and increased clinical validation, cfRNA is anticipated to emerge as a valuable tool in future clinical practice.

When analyzing circulating free RNA, the choice between using plasma or serum as the sample source significantly impacts research and diagnostic outcomes. Both have distinct characteristics affecting the analysis, composition, and handling. Plasma offers advantages in terms of its high circulating cell-free RNA (cfRNA) integrity due to the reduced RNA enzyme content resulting from the removal of cellular components like platelets during processing [2]. Furthermore, due to cell lysis, platelet activation, and various proteins and factors changing during coagulation, more nucleic acids are released, and the target cfRNA may be degraded, which can lead to more data noise and analytical complexity in serum compared to plasma [7,74]. cfRNA in plasma is more stable as it is isolated in the presence of an anticoagulant, minimizing coagulation. However, processing plasma can be complex, requiring the addition of anticoagulants that may complicate subsequent analysis steps. On the other hand, serum preparation is simpler as it does not necessitate anticoagulants and the serum can be separated via centrifugation post natural coagulation. However, serum poses a greater risk of cfRNA degradation due to the potentially higher RNA enzyme levels, which can impact analysis outcomes [12]. Alternatively, platelets may act as a carrier, carrying a certain amount of cfRNA and further decreasing the cfRNA in platelet-free serum during centrifugation. In conclusion, the choice between plasma and serum for cfRNA analyses hinges on various factors such as the sample handling, research objectives, and anticipated results, requiring careful consideration from investigators. The potential of cfRNA as a biomarker for disease diagnosis and prognosis has a wide-ranging impact on the future of clinical practice. The following are some potential future applications and directions for this promising biomarker. Early diagnosis and screening represent a key area where cfRNA holds great promise. By detecting subtle RNA changes, this biomarker can identify a patient’s risk prior to disease onset or the manifestation of symptoms, thus enabling earlier interventions and treatments. Personalized therapy is another valuable application of cfRNA analysis. By examining a patient’s RNA profile, healthcare professionals can tailor treatment plans, choose personalized and more effective strategies, and enhance treatment success rates. Additionally, cfRNA has the potential to monitor treatment responses in real time, facilitating timely adjustments to treatment plans, improving treatment outcomes, and minimizing treatment-related adverse events. In large-scale epidemiological studies, the analysis of cfRNA can be beneficial for simultaneously detecting and differentiating multiple diseases, given the shared molecular biomarkers across various diseases. The analysis of cfRNA plays a crucial role in advancing precision medicine, which can provide personalized prevention, diagnostic, and treatment options based on an individual’s genetic, physiological, and environmental characteristics [1]. A comprehensive understanding of cfRNA can improve the awareness of disease pathogeneses, disease progression patterns, and individual treatment responses, thereby optimizing the accuracy and effectiveness of healthcare. Specifically, the benefits of cfRNA analysis may include personalized diagnoses and treatment planning, the prediction of therapeutic responses and adverse effects, the promotion of targeted therapies and drug development, and the facilitation of interdisciplinary collaboration and data integration. Personalized diagnoses and treatment planning involve a closer examination of individual cfRNA profiles, allowing a deeper understanding of disease-specific biology [19]. This enables customizable diagnoses and treatment options in order to achieve improved therapeutic efficacy, unlike traditional medical methods that often follow a “one-size-fits-all” strategy, neglecting individual variations in genetic composition, physiological conditions, and environmental influences. Predicting therapeutic responses and adverse effects through changes in cfRNA levels can guide treatment decisions by indicating a patient’s responses to specific therapies, therapeutic effects, and potential adverse effects. This analysis can help in predicting patient responses to treatment modalities, optimizing treatment selection, and minimizing unnecessary treatments and adverse effects. cfRNA analysis supports targeted therapy and drug development by identifying molecular targets for diseases and offering insights for designing targeted therapeutic drugs to enhance the precision and efficacy of treatments. Evaluating drug efficacy and safety through cfRNA analysis accelerates the development and clinical application of new therapies. Furthermore, promoting interdisciplinary collaboration and data integration in cfRNA analysis is essential for realizing precision medicine. Collaboration across interdisciplinary fields such as biology, medicine, and bioinformatics is necessary, involving experts from different domains. Integrating various clinical, genomic, and epigenomic datasets establishes a robust data platform for precision medicine, facilitating personalized diagnostic and therapeutic strategies [129,130,131].

The wide adoption of cfRNA could serve as a valuable tool for these studies, offering insights into disease pathogenesis and influential factors. Continued technological advancements, including the use of high-throughput sequencing and bioinformatics tools, are likely to enhance the sensitivity and accuracy of cfRNA detection, addressing concerns around standardization and repeatability, and advancing its application in clinical practice. As clinical validation and trials progress, the integration of cfRNA with other diagnostic and monitoring methods may offer more comprehensive patient management. Ultimately, cfRNA is poised to be a pivotal tool in disease management, supporting precise and personalized medicine. However, the widespread utilization of cfRNA in clinical practice warrants addressing challenges such as standardizing ethical and regulatory protocols, an initiative that scientists and medical professionals are actively working on to realize its broader application.

## 9. Conclusions

cfRNA has garnered considerable attention regarding its potential for diagnosing, treating, and assessing the prognoses of diseases, positioning it to serve as a biomarker and regulator that can drive the advancement of precision medicine. Despite its promise, the complexity of cfRNA underscores the need for extensive and ongoing research to unravel its many unknown aspects and attain a comprehensive understanding of its biological function. Simultaneously, the continuous development in this field necessitates considering the sensitivity and specificity of detection technology to ensure its reliability in clinical applications. As clinical validation and trials progress, the combination of cfRNA with other diagnostic and monitoring methods holds promise for bolstering patient management and facilitating a more comprehensive approach to disease management, thereby supporting the progress of precision and personalized medicine. To enable the widespread clinical use of cfRNA, the challenges related to regulatory ethics and protocols must be addressed. Scientists and medical professionals are actively pursuing initiatives aimed at surmounting these challenges and expanding the implementation of cfRNA analysis methods in clinical practice. This review comprehensively discusses the role of cfRNA in various diseases and biological processes to establish a systematic framework of knowledge that can advance the utilization of cfRNA molecules as biomarkers or therapeutic targets by researchers.

## Figures and Tables

**Figure 1 diagnostics-14-01045-f001:**
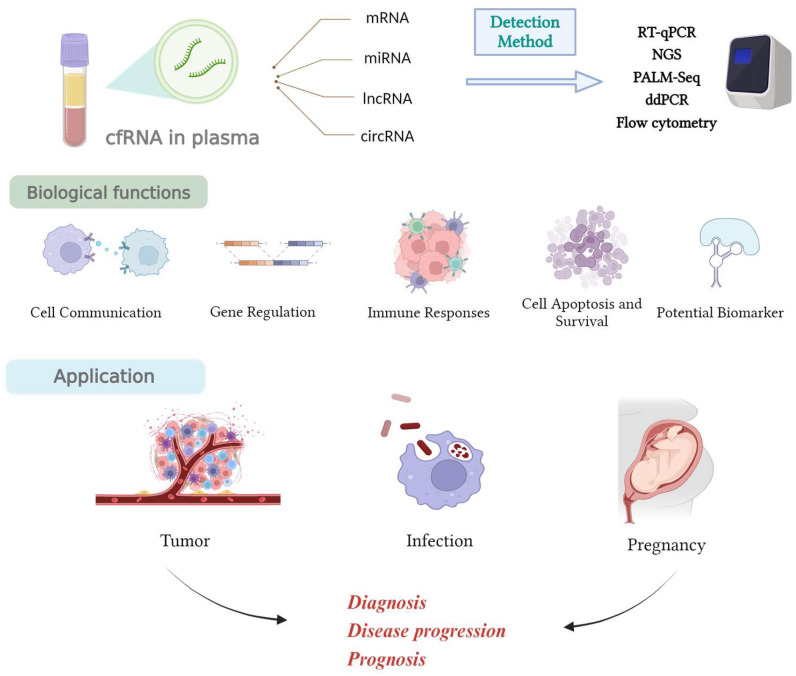
An overview of cfRNA in plasma for medicine. cfRNA in plasma includes various types of RNA such as miRNA, mRNA, lncRNA, and circRNA. Furthermore, various techniques like RT-qPCR have been developed to detect and analyze cfRNA in plasma, as we show in the first part of this paper. Also, cfRNA represents a complex and actively researched area within biology. These RNA molecules exist as fragments of varying lengths and are associated with multiple potential biological functions. Due to its abundant biological functions, this group of cfRNA could be applied in various diseases to diagnose patients, predict their progression, and assess their prognosis.

**Table 1 diagnostics-14-01045-t001:** The advantages and limitations of different methods for detecting cfRNA in plasma.

Detection Method		Advantages	Limitations
RT-qPCR	Real-time fluorescence Quantitative PCR	High sensitivity Good quantification Detects specific RNA sequences	Limited by unknown RNA sequences
NGS	Next-generation sequencing	High throughput Includes known and unknown sequences	High costs Data analysis difficulties
PALM-Seq	Polyadenylation ligation-mediated sequencing	Quantifies transcription Discovers new transcripts Variable splicing events RNA modifications	High costs Stringent RNA quality Integrity requirements Less gene-expression levels
ddPCR	Droplet digital PCR	Absolute quantification Detects a variety of RNA modifications Provides detailed RNA structure information	Complex instruments Professional techniques High costs Limited scope of application
Flow cytometry		Detects cell surface markers and proteins	Less suitable for complex RNA expression analyses

**Table 2 diagnostics-14-01045-t002:** Summary of detection methods for cfRNA in recent research.

Detection Method	Biomarker Type	Disease	Value	Reference
qRT-PCR	mRNA	HCC/MM and related pre-malignant diseases	Identification	[81]
	mRNA	Colorectal cancer	Monitoring progression	[78]
	mRNA	Preterm birth (PTB)	Predication	[82]
	mRNA	Thyroid cancer	Quantification of thyroid volume Recurrence predication	[80]
	mRNA	Glioma	Identification	[97]
	cfRNA	Preeclampsia	Predication	[98]
	mRNA	Pregnancy infections	Monitoring immune response and microbial infections during pregnancy	[99]
	mRNA	Melanoma	Identification of novel therapeutic targets or biomarkers	[100]
qRT-PCR and PCF RNA testing	mRNA miRNA	Embryonic trisomy 21 in the first trimester	Diagnosis	[79]
NGS	mRNA rRNA	Non-small-cell lung cancers	Monitoring progression Disease prognosis prediction	[101]
	Total cfRNA	Lung cancer	Early diagnosis	[102]
	cfRNA	Lung cancer	Diagnosis	[103]
NGS RT-ddPCR		Solid cancers	Early diagnosis	[76]
PALM-Seq	miRNA tRNA mRNA	COVID-19	Prediction	[84]
	mRNA miRNA	Preeclampsia	Predication	[83]
ddPCR	mRNA	Neuroblastoma	Diagnosis	[89]
	mRNA ctDNA	Pediatric solid tumors	Potential biomarker identification	[90]

## Data Availability

Not applicable.

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
