# Peer review of "A Comprehensive Review on Circulating cfRNA in Plasma: Implications for Disease Diagnosis and Beyond"

_diagnostics, 2024, doi:10.3390/diagnostics14101045_

Round 1

Reviewer 1 Report

Comments and Suggestions for Authors

The review article by Zhong and Bai et al focuses on plasma circulating RNA and their implications for diagnostics. The article is quite well written, covers aspects of methodology, potential applications and challenges. However, there are a few points which could be addressed and topics to be mentioned in this review, that in my opinion would improve the quality of this review.

Specific comments:

1.       On line 49 the authors write that cfRNA is more sensitive and “functional” than ctDNA. Could the authors elaborate on this? What do they exactly mean? The way this paragraph is written in its current form does not reflect the truth. For e.g., methylation patterns are widely documented in the literature to reflect tissue specific genomic information. While RNA profiles related to specific tissue sites or cell types are largely rely on deconvolution algorithms, thus specificity of cell-free RNA copies vs. methylated ctDNA copies is debatable. Moreover, when looking at oncogenic mutations which are detectable in cell-free (tumor) DNA (mutant ctDNA), a classification of cancerous vs. healthy origin can be made with high diagnostic accuracy given that oncogenic alterations are generally confined to tumor cell origins. Contrary to ctDNA, tumor-specific signals in cfRNA may be ‘diluted’ since many cell type i.e., including tumor and healthy cells can express the same gene. Moreover, tissue of origin predictions are based on gene expression signatures or deconvolution algorithms of RNA-Seq data, thus I doubt that functionality of cfRNA in this regards is superior to ctDNA.

2.       The authors should consider mentioning RNases and how their activity in blood hinders detection of e.g., mRNA.

3.       Keeping with the previous point, the authors should also consider mentioning challenges associated with the fragmented nature of circulating nucleic acids. How does this affect detection of certain RNA types by sequencing technologies (likelihood of recovering mRNA genes vs. miRNAs that are naturally shorter)?

4.       I’d appreciate to read an expanded section on isolation techniques/kits. Most kits capture total cell-free RNA including protein-bound, exosomal, miRNA and other small RNAs, others are specifically useful isolation of small RNAs like miRNA. There are many papers that show quality and type of RNA isolated greatly affects downstream signature detection (e.g., doi.org/10.1093/nsr/nwae022).

5.       Many researches debate whether plasma or serum is a better choice of blood analyte when it comes to cfRNA analysis. Of course in many cases availability of one or another determines the choice, still it would be useful for many readers to know what are the considerations when it comes to serum vs. plasma. Why plasma is a better choice than serum?

Comments on the Quality of English Language

Minor comments:

Spell check and English correction is recommended.

Author Response

Dear reviewer,

We are very appreciate your precious comment about our review“A Comprehensive Review on Circulating cfRNA in Plasma:Implications for Disease Diagnosis and Beyond

Please see the attachment for detail revision.

Reviewer 2 Report

Comments and Suggestions for Authors

This comprehensive review aimed to provid a detailed overview of the current knowledge and advancements in the study of plasma cfRNA, focusing on its diverse landscape and biological functions, detection methods, diagnostic and prognostic potential in various diseases, challenges, and future perspectives. The review was well witten, however I would suggest some improvements.

Please remove the background colour on Figure 1, give a more detailed explanation to it, and improve the image. I think it is a bit weak and superficial. The legend should be improved too.

Give a bit more perspective on the future of cfRNA in disease management. 

Author Response

(The authors gave the same response as above.)
